# Nuclear Export Inhibitor KPT-8602 Synergizes with PARP Inhibitors in Escalating Apoptosis in Castration Resistant Cancer Cells

**DOI:** 10.3390/ijms22136676

**Published:** 2021-06-22

**Authors:** Md. Hafiz Uddin, Yiwei Li, Husain Yar Khan, Irfana Muqbil, Amro Aboukameel, Rachel E. Sexton, Shriya Reddy, Yosef Landesman, Trinayan Kashyap, Asfar S. Azmi, Elisabeth I. Heath

**Affiliations:** 1Departments of Oncology, Karmanos Cancer Institute, Wayne State University School of Medicine, Detroit, MI 48201, USA; uddinh@wayne.edu (M.H.U.); liyi@karmanos.org (Y.L.); khanh@karmanos.org (H.Y.K.); kameelo@karmanos.org (A.A.); rachel.sexton@wayne.edu (R.E.S.); shriyar2003@gmail.com (S.R.); 2Department of Chemistry, University of Detroit Mercy, Detroit, MI 48221, USA; muqbilir@udmercy.edu; 3Karyopharm Therapeutics Inc., Newton, MA 02459, USA; ylandesman@karyopharm.com (Y.L.); trinayan@karyopharm.com (T.K.)

**Keywords:** nuclear export, eltanexor, KPT-8602, PARP inhibitors, castration resistance, prostate cancer

## Abstract

Aberrant nuclear protein transport, often observed in cancer, causes mislocalization-dependent inactivation of critical cellular proteins. Earlier we showed that overexpression of exportin 1 is linked to higher grade and Gleason score in metastatic castration resistant prostate cancer (mCRPC). We also showed that a selective inhibitor of nuclear export (SINE) selinexor and second generation eltanexor (KPT-8602) could suppress mCRPC growth, reduce androgen receptor (AR), and re-sensitize to androgen deprivation therapy. Here we evaluated the combination of KPT-8602 with PARP inhibitors (PARPi) olaparib, veliparib and rucaparib in 22rv1 mCRPC cells. KPT-8602 synergized with PARPi (CI < 1) at pharmacologically relevant concentrations. KPT-8602-PARPi showed superior induction of apoptosis compared to single agent treatment and caused up-regulation of pro-apoptotic genes *BAX*, *T**P53* and *CASPASE 9*. Mechanistically, KPT-8602-PARPi suppressed *AR*, *ARv7*, *PSA* and AR targets *FOXA1* and *UBE2C*. Western blot analysis revealed significant down-regulation of AR, ARv7, UBE2C, SAM68, FOXA1 and upregulation of cleaved PARP and cleaved CASPASE 3. KPT-8602 with or without olaparib was shown to reduce homologous recombination-regulated DNA damage response targets including *BRCA1*, *BRCA2*, *CHEK1*, *EXO1*, *BLM*, *RAD51*, *LIG1*, *XRCC3* and *RMI2*. Taken together, this study revealed the therapeutic potential of a novel combination of KPT-8602 and PARP inhibitors for the treatment of mCRPC.

## 1. Introduction

Metastatic prostate cancer (mPCa) is a lethal disease that remains the second leading cause of death among men in the United States [1]. For the past several decades Androgen deprivation therapy (ADT) has been considered as a primary treatment option for advanced symptomatic PCa [2]. Patients initially respond to ADT, however, most of them develop metastatic castration-resistant prostate cancer (mCRPC) [3]. Increasing evidence suggests that PCa and CRPC modulates androgen receptor (AR), Akt, Src, Wnt, hedgehog and other signal transduction pathways at the molecular level [4,5,6,7,8]. Growing evidence also reveals the involvement of growth factors such as transforming growth factors-β (TGF-β), insulin-like growth factors (IGFs) and epidermal growth factor (EGF) in the progression of CRPC [9,10]. Among these, the AR signaling is considered as the most important one [4]. Unfortunately, targeted small molecule inhibitors against AR such as abiraterone and enzalutamide modestly increase the overall survival by ~4–5 months [11]. In about 20% of patients, mutations in genes of the DNA damage repair (DDR) pathway such as *BRCA1*, *BRCA2*, *CHEK2*, *ATM*, *RAD51D*, and *PALB2* make the survival of the tumor dependent on the secondary DNA repair protein poly(ADP) ribose phosphate (PARP) [12,13]. These observations promoted the use of PARP inhibitors in a sub-group of mCRPC patients utilizing a synthetic lethal concept [14]. PARP inhibition can block the tumor growth via the inhibition of single-stranded DNA repair and trapping of PARP at the DNA damage site [15,16]. Despite promising pre-clinical and clinical studies and subsequent FDA approval, the overall outcome of PARP inhibitors such as olaparib has been modest in metastatic prostate cancer patients with *BRCA1* or *BRCA2* mutations. 

Macromolecules move across the nuclear membrane in a gated manner regulated by a set of transporters belonging to the karyopherin family [17]. This movement is essential for the proper functioning of proteins. For example, the ability to localize in the nucleus is essential for transcription factor activation, and spatial separation of proteins is commonly used as a mechanism for preventing spontaneous signal activation [18,19]. Exportin 1 (XPO1), also known as Chromosome Region Maintenance 1 (CRM1), is one of the seven mammalian export proteins that facilitates the proteins across the nuclear membrane to the cytoplasm through nuclear export signal (NES) recognition [20]. Aberrant over-expression and export activity, as observed in cancer, causes unusual efflux of tumor suppressor proteins (TSPs) to the cytosol leading to their functional inactivation [21,22]. 

Earlier we showed that increased XPO1 in prostate cancer is associated with a high Gleason score and bone metastasis [23]. Aberrant expression of tumor suppressors such as P63 in the cytoplasm is associated with increased prostate cancer-specific mortality up to 20 years after diagnosis [24]. The mislocalized expression was associated with reduced apoptosis and higher proliferative activity [24]. More significantly, BRCA1 and its network proteins carry NES and are bonafide nuclear export targets. This makes the restoration of tumor suppressor and genome surveillance protein in the nucleus through XPO1 inhibition an attractive therapeutic strategy. The current study focused on bringing forward a novel and effective combination simultaneously affecting AR and inhibiting DNA damage response (DDR) for mCRPC. BRCA1/2 deficient mCRPC growth cannot be sufficiently blocked by PARP inhibitors (PARPi) and AR blockers since the TSPs such as TP53, RB, FOXO, P27 are functionally inactivated due to their excessive export to the cytoplasm. 

Therefore, we hypothesize that compared to targeted therapeutics with narrow mechanisms of action, blocking the nuclear exporter XPO1 by SINE could have a broad spectrum activity on several tumor suppressor proteins, AR and AR splice variants, DDR elements, leading to better treatment outcomes in patients with mCRPC who are being treated with PARPi.

## 2. Results

### 2.1. KPT-8602 with PARP Inhibitors Synergistically Inhibits the Growth of Prostate Cancer Cells

In this work, we have used the 22rv1 mCRPC cell line which represents this disease and one of the very few available representative cell lines. Compared to non-mCRPC cell lines such as C4-2B or LNCaP, the basal level of XPO1 in 22rv1 cells was higher both at mRNA and protein levels. The 22rv1 cells also showed lower BRCA2 levels compared to non-mCRPC cell lines (Appendix A). We observed significant growth inhibition with SINE compound KPT-8602 with different PARP inhibitors, namely olaparib, veliparib and rucaparib (Figure 1A–C). Isobologram analysis revealed that the combination of KPT-8602 and PARP inhibitors exerted a synergistic inhibitory effect on 22rv1 cells (Figure 1A–C; right panel), as the combination index (CI) values were below 1 (Figure 1A–C; middle panel). To investigate whether this synergism is cell line dependent or not, we tested this combination on androgen independent prostate cancer cells C4-2B as well as androgen sensitive prostate cancer cells LNCaP. Interestingly, both cell lines showed synergy with KPT-8602 and PARP inhibitor olaparib, though the observed inhibitory concentrations were higher (Appendix A). To evaluate the growth inhibitory effect of KPT-8602 further, with or without PARP inhibitors, we performed clonogenic assays. At a 50 nM dose of KPT-8602, no significant changes were observed in the formation of colonies, however the growth inhibitory trends of KPT-8602 and PARPi combinations were reflected in the number of colonies that survived. A significantly lower number of colonies were observed in KPT-8602 and olaparib or rucaparib combination treatment groups (Figure 1D,E).

### 2.2. KPT-8602 Treatment with PARPi Enhanced Apoptotic Cell Death in 22rv1 Cells

To confirm the specific type of cell death we performed apoptosis detection assay. The flow cytometric analysis of annexin-V showed a significant increase in early apoptotic cell death in the KPT-8602 and olaparib combination group. Olaparib induced early apoptosis in 9.3% of cells (*p* = 0.0053) compared to control, and 15% (*p* = 0.0057) when combined with the SINE agent (Figure 2A,B). There was no significant difference between KPT-8602 and olaparib alone treatment (*p* = 0.183). A significant increase in KPT-8602-olaparib is indicative of synergy between these two compounds. We observed a similar trend in KPT-8602-veliparib and KPT-8602-rucaparib combinations (data not shown). To further confirm these findings, we checked the cleavage of PARP and CASPASE 3 proteins which are considered as markers of apoptosis. Western blot analysis demonstrated up-regulation of cleaved PARP and cleaved CASPASE 3 with KPT-8602-PARPi combination treatment (Figure 2C,D), in agreement with flow cytometry findings.

### 2.3. KPT-8602 and PARP Inhibitors Treatment Causes Upregulation of Apoptosis Related Genes in 22rv1 Cells

The nuclear export protein XPO1 is frequently overexpressed in cancers [25,26] and might be linked with apoptosis resistance. As a proof of concept, we overexpressed XPO1 in cancer cells and observed significant downregulation of *CASPASE 9* (Appendix A). In 22rv1 cells, to observe KPT-8602-PARPi effects on apoptosis-associated genes, we measured gene expression at 24 h using real-time RT-qPCR (Figure 3). Treatment with KPT-8602 alone caused ~1.5-fold upregulation of *BAX* and *T**P53* (Figure 3A,C) and ~1.2-fold up-regulation of *CASPASE 9* (Figure 3E) compared to untreated control. However, olaparib only treatment showed no change in the expression of *BAX* and *T**P53* but showed a slight down regulation of the *CASPASE 9* gene. On the other hand, we observed up-regulation of all three apoptosis-associated genes in veliparib and rucaparib treated groups either alone or in combination with KPT-8602 (Figure 3B,D,F). Rucaparib only treated cells showed about 2-fold more expression of *BAX* and *CASPASE 9* compared to untreated cells (Figure 3B,F), whereas combination with KPT-8602 caused 1.8-fold and 1.3-fold change, respectively, despite the earlier synergistic effect on growth inhibition and significant reduction in colony formation.

### 2.4. KPT-8602 Treatment with or without PARPi Down-Regulates AR and Its Target Genes

Treatment with the SINE compound KPT-8602 led to significant downregulation of androgen receptor (*AR*) and *ARv7*, an *AR* variant at the protein and mRNA levels (Figure 4A–C). Both AR and ARv7 are associated with prostate cancer resistance and metastasis [27,28]. The downstream target genes of *AR* including *FOXA1*, *UBE2C* were also reduced with KPT-8602. Most of the AR-associated genes remained unchanged with PARPi treatment. Interestingly, among PARPi, rucaparib alone caused significant down-regulation of *ARv7* (Figure 4C) and may play an important role in overcoming resistance.

The basal expression level of AR protein is very low in 22rv1 cells. Olaparib at 10 µM dose was shown to induce AR expression when used alone, but not in combination with KPT-8602 (Figure 4A(i,ii)). All of the PARP inhibitors showed downregulation of the AR variant (ARv) in combination with KPT-8602. FOXA1 was downregulated by both KPT-8602 and PARP inhibitors treatment either alone or in combination. AR signaling-associated protein SAM68 has been shown to be involved in a variety of cellular processes including alternative splicing and migration potential in various types of cancer [29,30]. We observed decreased expression of SAM68 in combination treatment (Figure 4A,B). A similar trend was also found in the case of prostate-specific antigen (PSA) (Figure 4A,B). 

Immunofluorescence analysis showed that baseline expressions of AR and ARv7 in 22rv1 cells are high and predominantly in the nucleus (Figure 5A,B). On the contrary, in normal HEK293 cells, baseline expression of AR and ARv7 is very low (Appendix A). KPT-8602 treatment with or without rucaparib causes slight nuclear accumulation of AR in 22rv1 cells, however, overall expression of AR in cells was reduced in the combination. Additionally, in combination treatment, not only cell number but also size of the nucleus gets reduced (Figure 5A). A similar effect was observed in the case of ARv7 expression by KPT-8602 and rucaparib treatment, however, rucaparib only treatment caused equal distribution of ARv7 throughout the cells (Figure 5B). As PARP inhibitors interfere with DNA damage pathways, we checked a number of DNA damage repair genes under KPT-8602 treatment with or without olaparib at 100 nM and 5 µM doses, respectively (Figure 5C). Interestingly, we observed significant downregulation of all genes tested *BRCA1*, *BRCA2*, *CHEK1*, *EXO1*, *BLM*, *RMI1*, *RAD54L*, *RAD51*, *LIG1*, *XRCC3* and *RMI2* with KPT-8602 alone or in combination with olaparib. The olaparib alone treatment was shown to reduce the expressions of *CHEK1* and *RAD51* but other gene expressions remain unchanged. The Cancer Genome Atlas (TCGA) data analysis revealed that the alteration frequency of the above mentioned genes was up to 5.3-fold in metastatic prostate adenocarcinoma compared to non-metastatic counterparts (Appendix A), suggesting an important role in metastasis. Moreover, STRING database analysis demonstrated that these genes are associated with AR signaling mostly through UBE2C (Appendix A). It has been shown experimentally that higher expression of UBE2C is regulated by BRCA1 [31]. These results implicate the role of *XPO1-AR-DDR* axis in overcoming castration resistance in prostate cancer.

## 3. Discussion 

In this study, we bring forward a novel combination of a new generation SINE compound, KPT-8602, with different PARP inhibitors for the first time. Importantly, KPT-8602 showed synergy with PARPi olaparib, veliparib and rucaparib at pharmacologically relevant concentrations in growth inhibition assay. KPT-8602-PARPi combinations triggered apoptosis. Mechanistically, KPT-8602-PARPi suppressed AR, ARv7 and the ARv7 downstream targets SAM68, FOXA1 and UBE2C. Excitingly, KPT-8602 with or without olaparib was able to reduce expression of DDR genes including *BRCA1*, *XRCC3*, *CHEK1* and *RAD51* that may play an important role in sensitizing mCRPC to the synthetic lethal action of PARPi.

The SINE compound KPT-8602 and PARPi have been individually tested in preclinical models of mCRPC [23,32] as well as in clinical settings (clinicaltrials.gov: NCT02649790, NCT02975934); however, their combination has never been evaluated. Preclinical data showed tumor growth inhibition in mCRPC with the first generation SINE compound selinexor (KPT-330) [23,26] and was tested in a phase II study (NCT02215161) in patients with mCRPC. Recently, a phase I/II study was performed to determine the safety, preliminary efficacy, and recommended phase II dose of KPT-8602 in abiraterone refractory mCRPC patients [33]. There is evidence that the second generation SINE compound KPT-8602 could have a better tolerability profile [34,35]. There are phase II (NCT02952534) and phase III (NCT02975934) clinical trials (TRION studies) ongoing with rucaparib for the treatment of mCRPC with HR deficiency. However, mCRPC patients that have activation of the HR pathway do not benefit from it. Therefore, new effective combinations are acutely needed. In the present study, we have observed synergy (CI < 1) between KPT-8602 and rucaparib as well as with other PARPi in *BRCA1* wild-type 22rv1 mCRPC cells. It needs to be mentioned that the 22rv1 cell line is one of the very few available representative cell lines of mCRPC which is androgen resistant. The cell line originated from a xenograft that was serially passaged in mice after castration-induced regression and relapse of the parental cells [36]. In 22rv1 cells, synergy is achieved with lower doses of KPT-8602, however, with higher doses of KPT-8602, synergy can be achieved even in a non-cancerous cell line. In HEK293 cells, the IC_50_ for KPT-8602 was 7-fold higher compared to 22rv1 cells but it was almost same for rucaparib, however, the combinations was synergistic (Appendix A). The several-fold higher IC_50_ values for KPT-8602, olaparib and rucaparib were observed against normal pancreatic islet cells (Appendix A). Besides cell growth inhibition, the colony formation ability of 22rv1 cells was also found to be significantly reduced with KPT-8602 and olaparib or rucaparib combinations, supporting the effectiveness of these combinations. 

Combination treatment led to enhanced apoptosis in most cases compared to treatment with the individual agents. Combination treatment further increases the expressions of pro-apoptotic genes *BAX*, *TP53* and *CASPASE 9* at the mRNA level, and cleaved PARP and CASPASE 3 at the protein level. KPT-8602 was also shown to induce apoptosis in mCRPC cells in our earlier study [23], however, the detailed mechanism involved in the preferential killing of cancer cells by KPT-8602 is currently unknown (Appendix A). One plausible mechanism can be the nuclear retention of NES bearing apoptosis inducers such as TP53, RB, FOXO etc. [37,38]. However, we did not observe any prominent change in TP53 accumulation in 22rv1 cells with short term (4 h) KPT-8602 treatment (Appendix A). On the other hand, PARP inhibitors target the DNA repair pathway to kill cancer cells preferentially [39]. In fact, olaparib has been shown to reduce tumor progression in some CRPC patients [40]. It has been revealed that PARPi enhances pre-existing DNA repair defects, which results in the accumulation of DNA double-strand breaks (DSB) and apoptosis [41]. Therefore, it is likely that a combination of KPT-8602 and PARPi can lead to increased apoptotic cell death of cancer cells in patients with mCRPC.

The effect of XPO1 on AR signaling inhibition is currently unknown. Among twenty known AR variants, AR and ARv7 are predominantly found in the nucleus; on the other hand, ARv1 and ARv6 remain mostly in the cytoplasm. To exert a proliferative signal, the full length AR and ARv7 are required to translocate ARv1 and ARv6 to the nucleus [42]. Hence, nuclear export inhibition could interrupt AR signaling with KPT-8602 even though it retains AR and ARv7 in the nucleus. One of the mechanisms of therapeutic failure in mCRPC is production of a constitutively active mRNA splice variant of *AR* (e.g., *ARv7*) [43,44]. Enhanced AR activity results in overexpression of *PSA* and a subsequent rise in the PSA level in the serum [45,46]. Activated AR also leads to overexpression of the mitotic phase (M-phase) regulatory gene *UBE2C* in CRPC [47]. In AR signaling, RNA-binding protein SAM68 and FOXA1 regulate AR splice variant expression (including *ARv7*) and upregulate *UBE2C* [48,49]. In this study, we observed downregulation of AR, ARv, PSA, SAM68, FOXA1 and UBE2C at the protein and mRNA levels after treatment with KPT-8602 and PARPi. This finding is in line with our previous report where we observed a similar impact through silencing XPO1 [23]. These results indicate that SINE-PARPi could become a superior combination that could potentially result in better clinical response in mCRPC patients with activated AR.

Androgen-targeted therapy is usually not effective in the treatment of CRPC. A study showed expression of a set of HR-associated genes in CRPC cells including *BRCA1*, *RMI2* and *RAD54L* [50]. Though the AR inhibitor enzalutamide was shown to create BRCAness in CRPC [50], it was not sufficient to suppress DDR genes or proteins. Furthermore, PARPi administration to *BRCA1/2* mutant cells blocks prostate cancer growth, however, several markers of homologous recombination (HR) signaling are activated, leading to modest clinical outcome [51,52]. Excitingly, we observed that KPT-8602 alone or in combination with olaparib caused a significant reduction in a set of DDR genes, including *BRCA1*, *CHEK1*, *RAD51*, *XRCC3*, and others. Similarly, in distinct solid tumor models, selinexor was shown to reduce the expression of DDR proteins [53]. XPO1 inhibition by selinexor was also shown to sensitize prostate cancer cells to docetaxel through a mechanism involving the enhancement of DNA damage [54]. Further, selinexor and analogs demonstrate antitumor effects through modulating the expression of *CYCLIN D1* and *SURVIVIN* in prostate cancer models [26]. SINE compounds were also shown to induce nuclear retention and stabilization of TP53 with a concomitant increase in γ-H2AX, indicative of double strand DNA breaks [55]. Moreover, preclinical studies demonstrated that AR signaling and DNA damage response (DDR) converge with several genes of HR under the regulation of the AR-DDR axis [50]. These results indicate that DNA damage response signaling could be impacted through nuclear export inhibition by KPT-8602 that warrants further exploration in the context of PARP inhibition in mCRPC. Collectively, our previously published results and current findings point to several signaling pathways including AR and DDR that can be targeted using KPT-8602-PARPi. A schematic diagram illustrating the current working hypothesis has been shown in Figure 6.

## 4. Materials and Methods

### 4.1. Cell Lines, Reagents and Antibodies

22rv1, LNCaP, VCaP (prostate cancer cell lines) and SNU-1 (gastric cancer cell line) were purchased from American Type Culture Collection (ATCC, Manassas, VA, USA). The human pancreatic islets of Langerhans cell line was purchased from AcceGen Biotech (catalog No. ABC-TC4286; Fairfield, NJ, USA). HEK293 cell line was obtained from Dr. Dirk Daelemans at KU Leuven University, Leuven, Belgium under a material transfer agreement. C4-2B cell line was obtained from Prof. Ping Dou, Wayne State University, Detroit, MI, USA. The 22rv1, LNCaP and SNU-1 cell lines were maintained in RPMI-1640 medium (1×; Gibco, Waltham, MA, USA; catalog No. 11875-093), C4-2B in modified DMEM/F12 (1×; Gibco, Waltham, MA, USA; catalog No. 11320-033) and VCaP as well as HEK293 and islet cell lines in DMEM (1×; Gibco, Waltham, MA, USA; catalog No. 11965-092) media. All media were supplemented with 10% fetal bovine serum (FBS) (Sigma-Aldrich, St. Louis, MO, USA; catalog No. F0926), 100 U/mL penicillin and 100 μg/mL streptomycin (GE Healthcare, Chicago, IL, USA; catalog No. SV30010). The cell lines were maintained in a humidified incubator with 5% CO_2_ atmosphere at 37 °C. The cell lines have been tested and authenticated in a core facility of the Applied Genomics Technology Center at Wayne State University. The method used for testing was short tandem repeat (STR) profiling using the PowerPlex^®^ 16 System from Promega (Madison, WI, USA). SINE compound KPT-8602 (Karyopharm Therapeutics, Newton, MA, USA) was dissolved in dimethyl sulfoxide (DMSO). For western blotting analysis, Anti-AR (N20, catalog No. sc-816; which recognizes both AR full length and AR splice variants), anti-PSA (catalog No. sc-7316), anti-SAM68 (catalog No. sc-1238), anti-XPO1 (catalog No. sc-74454), anti-BRCA2 (catalog No. sc-28235) and anti-GAPDH (catalog No. sc-365062) primary antibodies were purchased from Santa Cruz Biotechnology (Santa Cruz, Dallas, TX, USA). Anti-FOXA1 (catalog No. NB-100-1262) was purchased from Novus Biologicals (Littleton, CO, USA). Anti-PARP (Cell Signaling Technology, Danvers, MA, USA; catalog No. 9542) and anti-CASPASE 3 (detect both full length CASPASE 3 and Cleaved CASPASE 3) were purchased from Cell Signaling Technology (Danvers, MA, USA; catalog No. 9662). Anti-β-ACTIN was purchased from Sigma (St. Louis, MO, USA; catalog No. A2228). For immunofluorescence analysis, Anti-AR (catalog No. 5153) and Anti-TP53 (catalog No. 2524) antibodies were purchased from Cell Signaling Technology (Danvers, MA, USA). Anti-ARv7 (catalog No. 31-1190-00) antibody was purchased from RevMAb Biosciences (South San Francisco, CA, USA). 

### 4.2. Growth Inhibition Assay

A total of 6000 22rv1 cells were seeded per well in 96-well plates. Following attachment, cells were treated with KPT-8602 and PARP inhibitors at varying doses as indicated for 72 h. Growth inhibition was determined by MTT [3-(4,5-dimethylthiazol-2-yl)-2,5-diphenyltetrazolium bromide] (Sigma-Aldrich, St. Louis, MO, USA; catalog No. M2128) assay. The MTT solution was added to the media at a final concentration of 0.8 mg/mL and cells were incubated at 37 °C for 2 h. After aspiration of media, formazan crystals were dissolved in DMSO. Optical densities were measured at 570 nm using SynergyHT plate reader (BioTek, Winooski, WI, USA). To calculate IC_50_ values for all drugs, GraphPad Prism Software were used (GraphPad Software, San Diego, CA, USA). Combination index (CI) was calculated and an isobologram was generated using Calcusyn 2.1 software (Biosoft, Cambridge, UK).

### 4.3. Colony Formation Assay

A total of 500 cells per well were seeded in the 6-well culture dishes at least in triplicate. Following treatment for 72 h, cells were maintained for 10 days in drug-free medium. Colonies were washed in PBS and fixed in 100% methanol. Colonies were stained with 0.5% crystal violet for 20 min and washed in tap water. After air drying, colonies containing more than 50 cells were counted.

### 4.4. Apoptosis Assay

The apoptotic death of cells was determined using Annexin V-FITC apoptosis detection kit (Biovision, Danvers, MA, USA) according to the manufacturer’s protocol. 22rv1 cells were treated with 100–300 nM KPT-8602 for 72–96 h. At the end of treatment, cells were trypsinized and stained with annexin V-FITC and propidium iodide (PI). The stained cells were analyzed using a Becton Dickinson flow cytometer at the Karmanos Cancer Institute Flow Cytometry Core within 1 h of staining.

### 4.5. Real-Time RT-qPCR

Total RNAs were extracted using the RNeasy Mini Kit (QIAGEN, Valencia, CA, USA) following manufacturer’s protocol. Complementary DNA was prepared using High Capacity cDNA Reverse Transcription Kit (Applied Biosystems, Waltham, MA, USA). Gene expression was detected by SYBR Green using StepOnePlus real-time PCR system (Applied Biosystems, Waltham, MA, USA). The sequences of primers used in this study are listed in Table 1. The SYBR Green master mix was activated by heating at 95 °C 10 min. A total of 40 thermal cycles were used consisting of 15 s at 95 °C and 1 min at 60 °C steps followed by melting curve analysis. Data were analyzed according to the comparative 2^−∆∆Ct^ method and were normalized to *ACTIN*, *GAPDH* and/or *18S* rRNA expression.

### 4.6. Overexpression of XPO1

The mammalian XPO1 (CRM1) (NM_003400) expression vector (pCMV6-CRM1; human-tagged ORF clone) (catalog No. RC206004) and empty vector (pCMV6-Entry) (catalog No. PS100001) were purchased from OriGene (Rockville, MD, USA). They were transfected to SNU-1 cancer cells using Lipofectamine 3000 transfection system (Invitrogen, Carlsbad, CA, USA) according to the manufacturer’s instruction. Briefly, after trypsinization cells were resuspended in the culture medium and seeded at 5 × 10^5^ cells/well in a 6-well plate. The next day, cells were transfected with XPO1 sequence containing vector or empty vector in serum-free medium. Expression was confirmed after 48 h of transfection by Real-time RT-qPCR.

### 4.7. Immunofluorescence Analysis

About 5000–30,000 cells were cultured on 8-chamber slides and exposed to SINE compound and/or PARP inhibitor for 4 h or 24 h. After treatment, 4% paraformaldehyde was used to fix the cells for about 10 min. Cells were permeabilized in 0.05% Triton-X100 for 15 min. After three times washing with 1× PBS, blocking was done with 5% bovine serum albumin (BSA) prepared in PBS containing 0.1% tween-20 (PBST). Cells were incubated for an hour and subjected to primary antibody (1:100 to 1:2000 dilution in 3% BSA) incubation for overnight at 4 °C. After primary antibody staining, cells were washed with PBST followed by PBS only washing. The goat anti-rabbit secondary antibody Alexa Fluor^®^ 488 conjugate (Invitrogen, Waltham, MA, USA; catalog No. A-11008) or goat anti-mouse secondary antibody Alexa Fluor^®^ 647 conjugate (Invitrogen, Waltham, MA, USA; catalog No. A-21236) was used as a secondary antibody (1:400 dilution in 3% BSA). Nuclear stain (DAPI) containing ProLong Gold antifade reagent (Thermo Fisher Scientific, Waltham, MA, USA) was added to the slides. A cover slip was placed over the slide and sealed with nail polish. An inverted fluorescent microscope (EVOS; Thermo Fisher Scientific, Waltham, MA, USA) was used to capture images at 40X resolution. For cellular aggregate staining, initial steps were performed in microcentrifuge tubes and placed on slides just before nuclear staining.

### 4.8. Western Blot Analysis

The total proteins were isolated using RIPA buffer (10×; Cell signaling technology, Danvers, MA, USA; catalog No. 9806). Protein concentration was measured using BCA Protein Assay (Pierce Biotechnology, Waltham, MA, USA) reagents. A total of 40 µg of proteins were loaded and subjected to 10% or 15% SDS-PAGE. The protein samples were transferred on nitrocellulose membranes and incubated with 1:1000 dilution of specific primary antibodies, and subsequently incubated with HRP-conjugated secondary antibodies (Santa Cruz Biotechnology, Santa Cruz, CA, USA). The signal was detected using the chemiluminescent detection system (Pierce Biotechnology, Waltham, MA, USA). For the determination of band density, NIH ImageJ 1.5Oi software was utilized. To calculate the relative level of protein expressions, band densities were normalized against ACTIN or GAPDH.

### 4.9. Statistical Analysis

The experiments were done at least in triplicate. Data are represented as mean ± standard deviation (SD). The data were compared using Student’s t-test. The *p* value of < 0.05 was considered as statistically significant. Symbol * indicates *p* < 0.05, ** indicates *p* < 0.01 and *** indicates *p* < 0.001.

## 5. Conclusions

Taken together, this study revealed the therapeutic potential of a novel combination including second generation SINE compound KPT-8602 and PARP inhibitors for the growth inhibition of mCRPC cells. XPO1 is a viable therapeutic target in mCRPC and KPT-8602-PARPi may exert its effect by suppressing AR signaling and DDR pathways. These findings warrant further preclinical and clinical validation of KPT-8602-PARPi combinations in mCRPC.

## Figures and Tables

**Figure 1 ijms-22-06676-f001:**
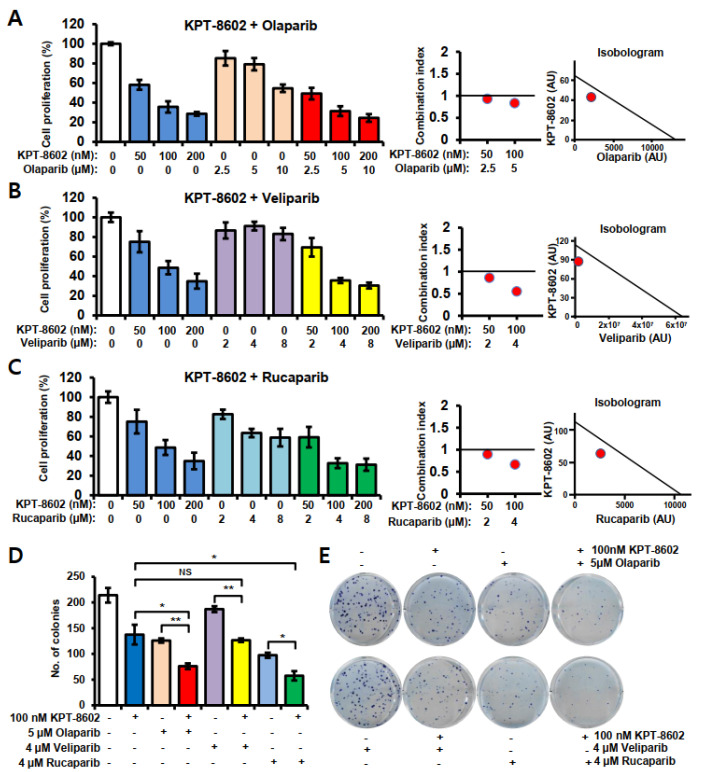
KPT-8602 with PARP inhibitors induced growth inhibition of 22rv1 prostate cancer cells. (**A**–**C**) 22rv1 cells were treated with 50–200 nM doses of KPT-8602 or 2–10 µM doses of olaparib, veliparib and rucaparib for 72 h. MTT assay was performed to determine the growth inhibition. Left panel is the growth inhibition bar diagram with single or combinations of KPT-8602 and PARP inhibitors. Middle and right panels are the corresponding combination indexes and isobolograms of A–C, respectively, which were generated using Calcusyn 2.1 software. (**D**–**E**) Colony formation with single or combinations of KPT-8602 and PARP inhibitors. D. Bar diagram showing the number of colonies with indicated doses of KPT-8602 or PARP inhibitors. E. Representative plates showing colonies with either alone or a combination of KPT-8602 and PARP inhibitors. All experiments have been done at least thrice. Error bars represent standard deviation. NS, not significant; * *p* < 0.05; ** *p* < 0.01.

**Figure 2 ijms-22-06676-f002:**
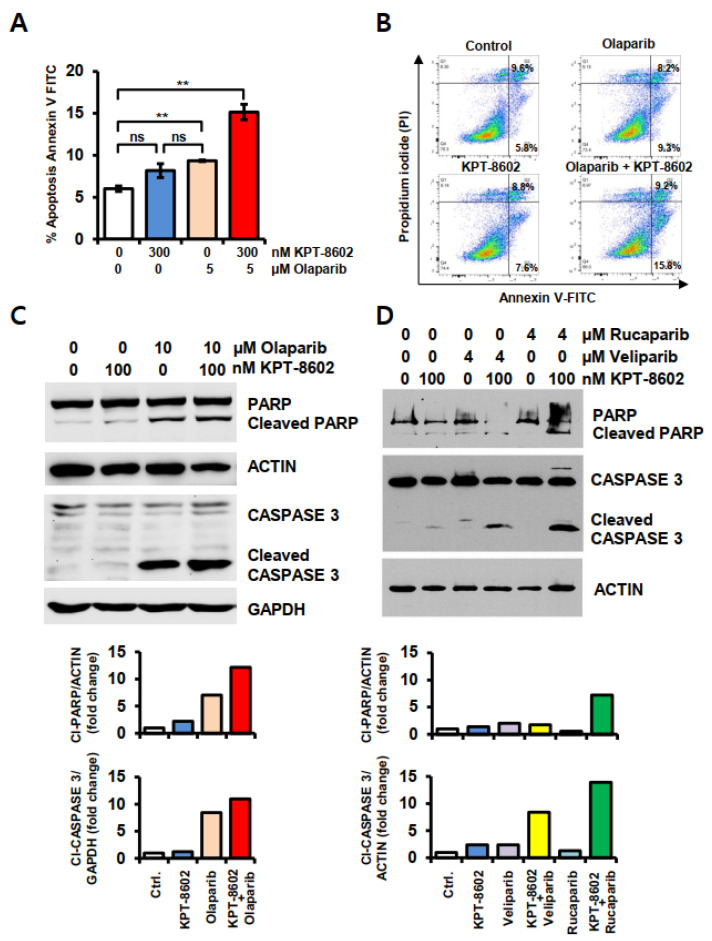
Apoptotic cell death with KPT-8602 and PARP inhibitors in 22rv1 cells. Apoptosis was determined by the flow cytometric detection of annexin-V and propidium iodide (PI) and western blotting technique. (**A**) Apoptosis in KPT-8602 and olaparib treated cells. (**B**) Representative flow cytometric image of KPT-8602 and olaparib-treated cells. (**C**) Western blot analysis of PARP and CASPASE 3 expression in KPT-8602 and olaparib-treated cells and (**D**) Western blot analysis of PARP and CASPASE 3 expression in KPT-8602 and veliparib- or rucaparib-treated cells. ns = not significant; ** *p* < 0.01. Lower panel of **C** and **D** shows relative protein expression obtained from band densities. NIH ImageJ 1.5Oi software was used to determine band density.

**Figure 3 ijms-22-06676-f003:**
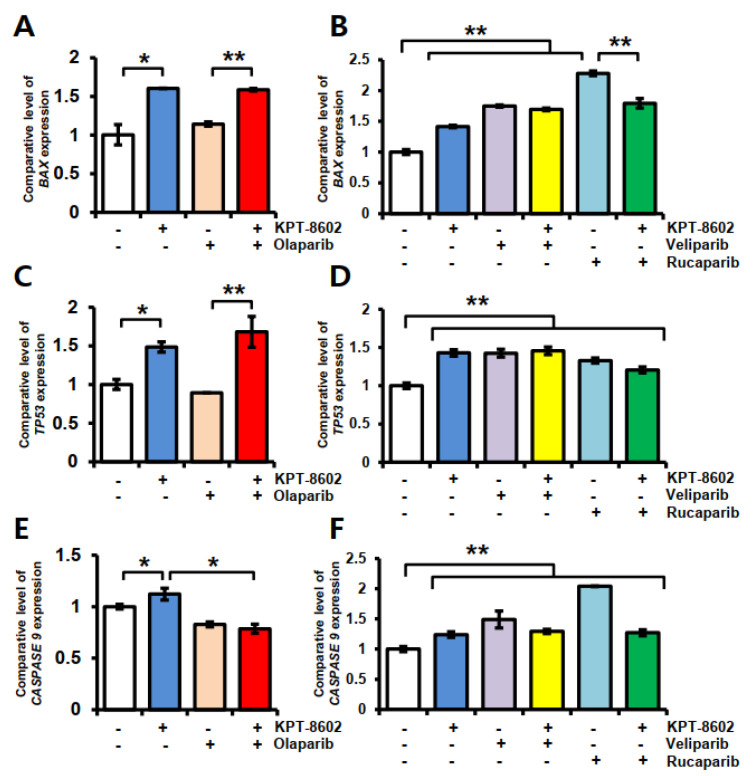
Expression of apoptosis-associated genes in response to KPT-8602 and PARP inhibitors treatment in 22rv1 cells. Gene expression was determined after 24 h of treatment using real-time RT-qPCR. (**A**,**B**) mRNA expression of *BAX*; (**C**,**D**) mRNA expression of *TP53* and (**E**,**F**) mRNA expression of *CASPASE 9*. All experiments have been done at least three times. Error bars represent standard deviation. * *p* < 0.05; ** *p* < 0.01.

**Figure 4 ijms-22-06676-f004:**
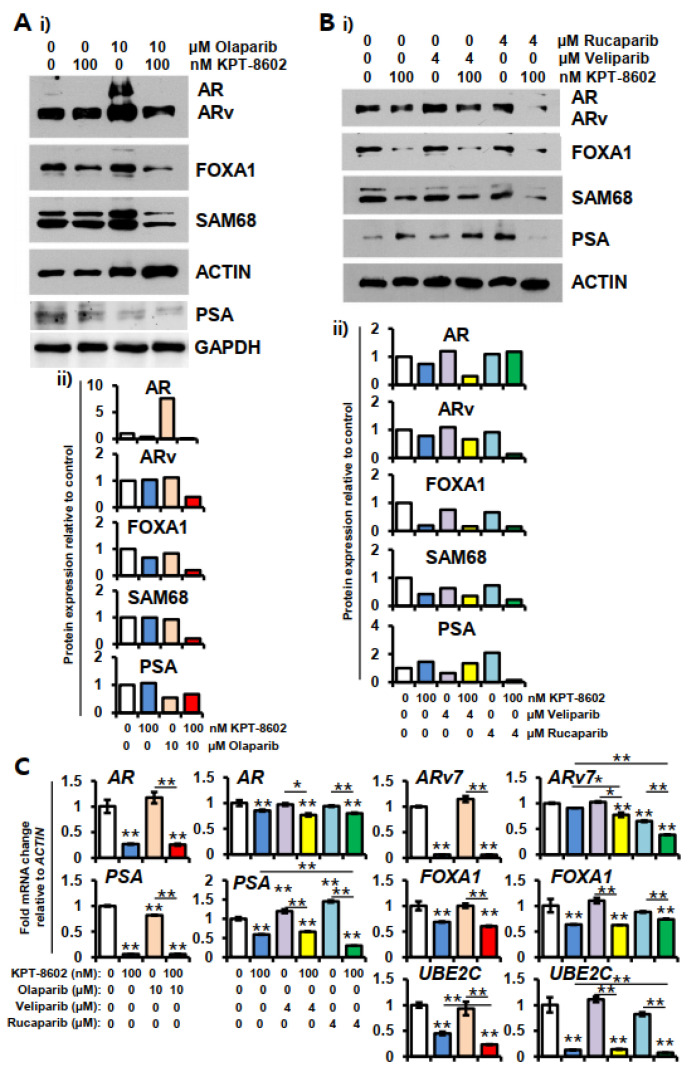
Expression of androgen receptor (AR) and its downstream target proteins and genes in 22rv1 cells treated with KPT-8602 and PARP inhibitors. Protein expression was determined by western blot analysis after 72 h of treatment and gene expression was determined after 24 h of treatment using real-time RT-qPCR. (**A**) Expression of AR, ARv, FOXA1, SAM68 and PSA in KPT-8602 and olaparib treated cells. (**B**) Expression of AR, ARv, FOXA1, PSA and SAM68 in KPT-8602 and veliparib- or rucaparib-treated cells. Lower panel shows relative protein expression obtained from band densities. NIH ImageJ 1.5Oi software was used to determine band density. (**C**) The mRNA expression of *AR*; *ARv7*, *PSA*, *FOXA1* and *UBE2C* under specified treatment conditions. The color codes of designated treatment are consistent among the groups. All experiments have been done in triplicate and error bars represent standard deviation. * *p* < 0.05; ** *p* < 0.01.

**Figure 5 ijms-22-06676-f005:**
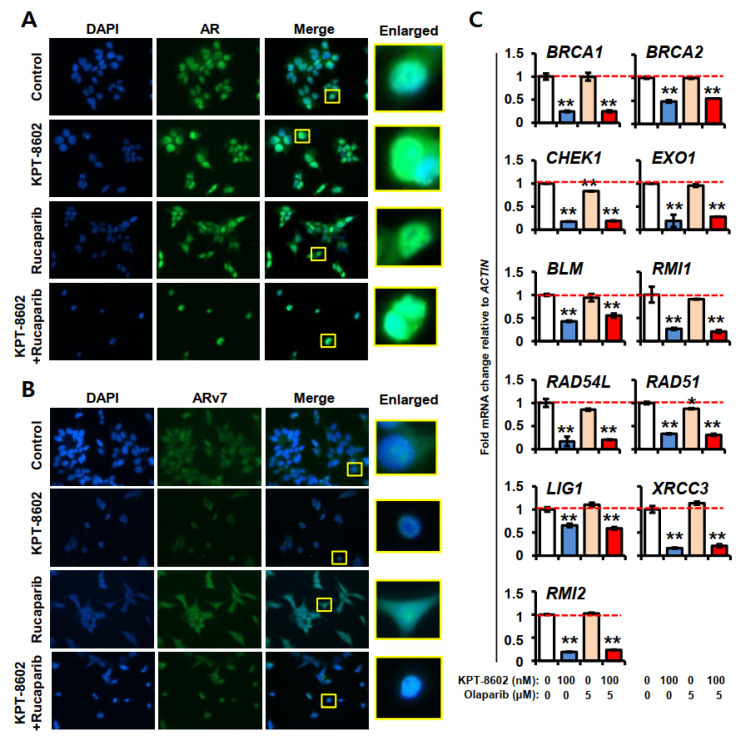
AR and ARv7 localization and DNA damage response of prostate cancer cells under KPT-8602 and PARP inhibitors treatment. (**A**,**B**) Immunofluorescence (IF) analysis of AR and ARv7 in 22rv1 cells. Cells were grown in cell culture-treated chamber slides and treated with 2 µM dose of KPT-8602 and 10 µM dose of rucaparib alone or in combination. After 24h cells were fixed, permeabilized and stained with 1:600 and 1:100 dilution of AR and ARv7 (green) antibodies, respectively. Nuclei were stained with DAPI (blue). (**C**) Gene expression was determined after 24 h of treatment using real-time RT-qPCR. mRNA expressions of *BRCA1*, *BRCA2*, *CHEK1*, *EXO1*, *BLM*, *RMI1*, *RAD54L*, *RAD51*, *LIG1*, *XRCC3* and *RMI2* in KPT-8602 and/or olaparib treated cells. Red dotted line indicates the expression level of the control. Doses for KPT-8602 and olaparib were 100 nM and 5 µM, respectively. Experiments have been done in triplicate. Error bars represent standard deviation. * *p* < 0.05; ** *p* < 0.01.

**Figure 6 ijms-22-06676-f006:**
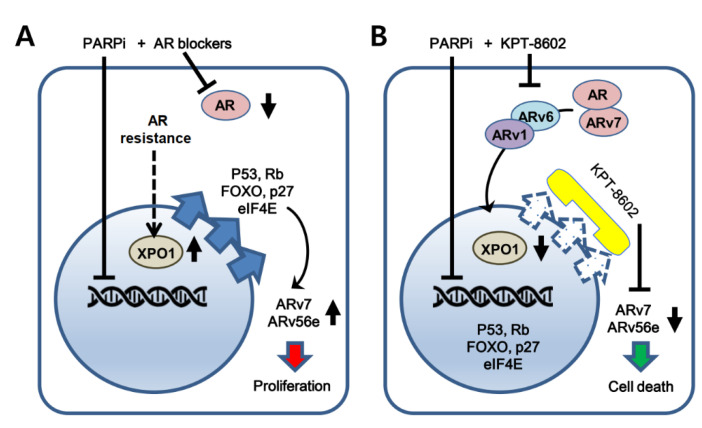
Schematic diagram illustrating the working hypothesis for the sensitization of mCRPC cells to XPO1 and PARP inhibitors. (**A**) Mechanism of drug resistance with traditional PARP inhibitor and AR-targeted agent treatment. (**B**) Sensitization of mCRPC cells using second generation SINE compound KPT-8602 and PARP inhibitors. BRCA1/2 deficient mCRPC growth cannot be sufficiently blocked by PARP inhibitors and AR blockers since the tumor suppressor proteins (TSPs) are functionally inactivated due to their excessive export to the cytoplasm. KPT-8602 can retain TSPs such as TP53, RB, FOXO, P27 and AR translation promoter eIF4e in the nucleus to facilitate the induction of cell death signals. Moreover, by keeping AR and ARv7 in the nucleus, KPT-8602 interrupts AR- and ARv7-mediated import of ARv1 and ARv6 in the nucleus.

**Table 1 ijms-22-06676-t001:** List of primers and sequences used for RT-qPCR.

Name of the Primers	Direction	Sequences (5′-3′)	References
*AR*	Forward	GACTTCACCGCACCTGATG	Aboukameel et al. 2018 [23]
	Reverse	AATGGGCAAAACATGGTCCC	
*ARv7*	Forward	TGTCCATCTTGTCGTCTTCGG	
	Reverse	TGCAATTGCCAACCCGGAAT	
*PSA*	Forward	GTCCCGGTTGTCTTCCTCAC	
	Reverse	CTCCCACAATCCGAGACAGG	
*FOXA1*	Forward	ACCAGCGACTGGAACAGCTA	
	Reverse	GTCATGTTGCCGCTCGTAGT	
*UBE2C*	Forward	TCCTGTCTCTCTGCCAACGC	
	Reverse	TTGTCTGATTCAGGGAAGGCA	
*ACTIN*	Forward	GCACAGAGCCTCGCCTT	
	Reverse	TCATCATCCATGGTGAGCTG	
*18 S*	Forward	GCAATTATTCCCCATGAACG	
	Reverse	GGCCTCACTAAACCATCCAA	
*BRCA1*	Forward	GGAACCTGTCTCCACAAAGTGT	This study
	Reverse	ACCTGTGTCAAGCTGAAAAGC	This study
*BRCA2*	Forward	AGTTCCCTCTGCGTGTTCTC	This study
	Reverse	GGGTATGAGCCATCCACCAT	This study
*BLM*	Forward	GAGTCTGCGTGCGAGGATTA	This study
	Reverse	CAGGTGTTTTTGCTACTGACACA	This study
*XRCC3*	Forward	GAAGAGGAGTGCGGAACCC	This study
	Reverse	CTGTGCACATCCTGCTGAGA	This study
*EXO1*	Forward	TCCATTGTGAAAAGACCAAGAAGTG	This study
	Reverse	CCATTTACCAGGTCAGGCAC	This study
*RMI1*	Forward	GCGGCGGTTCCTGTCCTTA	This study
	Reverse	TTGAAACCTCCACTGCTCAGAA	This study
*RMI2*	Forward	CAGGGTAGTGATGGCGGACC	This study
	Reverse	CCACTCCCATCACCATCACAT	This study
*RAD54L*	Forward	TGGTCCTACACTCTTAGCCG	This study
	Reverse	TCTCACTGCTGGATTTCCGT	This study
*CHEK1*	Forward	CAGTGGTGGGCAAAGGACAGT	This study
	Reverse	GTCTACGGCACGCTTCATATCT	This study
*LIG1*	Forward	CGAAGAAAAGTGCTGGACAGG	This study
	Reverse	TTTACCCTCTTTCTTGGGGTGG	This study
*RAD51*	Forward	GCTGGGAACTGCAACTCATCT	This study
	Reverse	GCTGCATCTGCATTGCCATTA	This study
*GAPDH*	Forward	CCACATCGCTCAGACACCAT	This study
	Reverse	ACCAGAGTTAAAAGCAGCCCT	This study
*XPO1*	Forward	ACGAGGAAGGAAGGAGCAGT	This study
	Reverse	CGAGCTGCATGGTCTGCTAA	This study
*CASPASE 9*	Forward	TGTTCAGGCCCCATATGATCG	This study
	Reverse	CAACTTTGCTGCTTGCCTGT	This study
*BAX*	Forward	AGGTCTTTTTCCGAGTGGCA	This study
	Reverse	CCCGGAGGAAGTCCAATGTC	This study
*BCL2*	Forward	CCTGGCTGTCTCTGAAGACTC	This study
	Reverse	GGGGCAGGCATGTTGACTTC	This study
*TP53*	Forward	TGACACGCTTCCCTGGATTG	This study
	Reverse	TTTTCAGGAAGTAGTTTCCATAGGT	This study

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
