# Peer review of "Nuclear Export Inhibitor KPT-8602 Synergizes with PARP Inhibitors in Escalating Apoptosis in Castration Resistant Cancer Cells"

_ijms, 2021, doi:10.3390/ijms22136676_

Round 1

Reviewer 1 Report

This is a nicely written research manuscript where the authors reported that a nuclear transport inhibitor (KPT-8602) could be instrumental in combination with PARP inhibitors for the treatment of mCRPC. They performed western, MTT, apoptosis, colony formation, RT-qPCR, and immunofluorescence assays to validate their results. The data were nicely presented and the results support their hypothesis. It would be really good if the similar experiments could be performed in mouse models later. 

Taken together, this study revealed the therapeutic potential of a novel 28
combination of KPT-8602 and PARP inhibitors for the treatment of mCRPC. 

Author Response

This is a nicely written research manuscript where the authors reported that a nuclear transport inhibitor (KPT-8602) could be instrumental in combination with PARP inhibitors for the treatment of mCRPC. They performed western, MTT, apoptosis, colony formation, RT-qPCR, and immunofluorescence assays to validate their results. The data were nicely presented and the results support their hypothesis. It would be really good if the similar experiments could be performed in mouse models later.

Taken together, this study revealed the therapeutic potential of a novel 28

combination of KPT-8602 and PARP inhibitors for the treatment of mCRP

-Response: Thank you for your comments. We have planned the in vivo experiment to assess the best in vitro combinations of KPT-8602 and PARP inhibitors in mCRPC xenografts. We anticipate to begin our animal studies after full restoration of animal core post covid restrictions.

Reviewer 2 Report

  1. The research article describing effects of KPT-8602 synergizes with PARP inhibitors in castration resistant metastatic prostate cancer has now been reviewed.
  2. There are concerns with the current manuscript starting with the title. Why have the authors boldly mentioned castration resistant PC with in vitro study? it should be corrected to "castration resistant cells".
  3. Furthermore, the title should be made more specific with the studies performed. Ex: Nuclear export inhibitor KPT-8602 synergizes with PARP inhibitors in escalating apoptosis in castration resistant cells
  4. The introduction is not sufficient. There is no briefing about insights into castration resistant PC, PARP inhibition etc.
  5. What was the use of human pancreatic cells in the current study? Why was it included in the methods?
  6. Result section "KPT-8602 and PARP inhibitors treatment causes upregulation of apoptosis related genes in 22rv1 cells" is unclear. Have the authors observed escalation of apoptosis in cells overexpressed with XPO1 when treated with KPT-8602? 
  7. In the same section, authors need to compulsorily show the KPT-8602 alone treated group similar to Rucaparib only treated cells since it is the main treatment in the current study. 
  8. The study design is fair but authors are advised to include pathways underlying CRPC like "Src, Wnt, growth factor mediated pathways etc." to add more rationale to the current study.
  9. Authors failed to provide color legend for figure3. Figure 5 needs to be improved as the nuclear accumulation is not visible. Further, the figure 5 A and 5B required quantification to clearly enumerate results to the readers. 
  10. Why have not the authors included a group without XPO1 treatment to observe the effects of differences in caspases?
  11. The current study contains flaws and needs extensive attention in order to turn it into a better refined study outcome. 

Round 2

Reviewer 2 Report

The manuscript is now in an acceptable form.